# Fluoride Occurrence and Human Health Risk in Drinking Water Wells from Southern Edge of Chinese Loess Plateau

**DOI:** 10.3390/ijerph16101683

**Published:** 2019-05-14

**Authors:** Hui Jia, Hui Qian, Wengang Qu, Le Zheng, Wenwen Feng, Wenhao Ren

**Affiliations:** 1School of Environmental Science and Engineering, Chang’an University, No.126 Yanta Road, Xi’an 710054, China; jiahui@chd.edu.cn (H.J.); wengang_qu@163.com (W.Q.); zhengle@chd.edu.cn (L.Z.); fww@chd.edu.cn (W.F.); 2018renwenhao@sina.com (W.R.); 2Key Laboratory of Subsurface Hydrology and Ecological Effect in Arid Region of Ministry of Education, Chang’an University, No.126 Yanta Road, Xi’an 710054, China

**Keywords:** high fluoride groundwater, drinking water safety, mechanism, hydrogeochemistry, human health risk assessment, loess aquifer

## Abstract

Fluoride hydrogeochemistry and associated human health risks implications are investigated in several aquifers along the southern edge of the Chinese Loess Plateau. Locally, 64% shallow groundwater samples in loess aquifer exceed the fluoride limit (1.5 mg/L) with the maximum of 3.8 mg/L. Presently, the shallow groundwater is the main source of private wells for domestic use, and this is clearly a potential risk for human health. Hydrogeochemistry and stable isotopes are used to elucidate the diversity of occurrence mechanisms. Enrichment of fluoride in groundwater is largely controlled by the F-containing minerals dissolution. Furthermore, alkaline condition and calcium-removing processes promote water–rock interactions. Stable isotopes of hydrogen and oxygen (δD and δ^18^O) in study area waters demonstrate that groundwater in loess aquifer is old, which means groundwater remains in the aquifer for a long time. Long residence time induces sufficient water–rock interactions, which play significant roles in the resolution of fluoride minerals. Samples from the shallow loess aquifer show elevated fluoride levels, which may pose human health risk for both adults (60%) and children (94%) via oral intake. To ensure drinking water safety, management measures such as popularizing fluoride-removing techniques and optimizing water supply strategies need to be implemented.

## 1. Introduction

Groundwater has been extensively developed and utilized as a source of drinking water due to its good quality [1]. Our demand for clean and safe drinking water requires strategies to utilize groundwater effectively, safely, and economically. According to the WHO guideline, fluoride concentrations between 0.5 to 1.5 mg/L provide optimum benefits to human health [2]. Exposure to high fluoride through drink water has been linked to endemic fluorosis [3,4]. 

Groundwaters containing elevated levels of fluoride have been reported on almost every continent, from Syria through Jordan, Egypt, Libya, Algeria, Sudan, and Kenya, and from Turkey through Iraq, Iran, Afghanistan, India, Sri Lanka, northern Thailand, and China. High fluoride groundwater is also observed in America, Canada, and Australia [5,6,7,8,9,10,11,12]. Fluoride enters the human body primarily through drinking water. Therefore, high fluoride risk in some tropical regions like Ghana is caused by high per capita water consumption [13]. Factors associated with endemic fluorosis include both hydrogeochemistry and social economy [14]. 

Managing the groundwater fluoride problem is a constant challenge. Effective management requires a knowledge of the fluoride distribution together with a full understanding of the potential sources. Many groups have investigated the mechanisms and geochemical processes regarding the occurrence of fluoride in groundwater. Fluoride occurs in granitoids, metamorphic rocks, igneous, and sedimentary rocks [15]. In most cases, enrichment of fluoride in groundwater is largely associated with the dissolution of F-bearing rocks. The occurrence of fluoride-containing minerals in these rocks is similar, such as fluorite (CaF_2_), biotite (K_3_AlSi_3_O_10_(F,OH)_2_), sellaite (MgF_2_), cryolite (Na_3_AlF_6_), mica, clays, and phosphorite [16]. The dissolution rate of fluoride minerals is controlled by the geological environment. Fluoride ions in minerals can be replaced by hydroxide ions due to similarities in charge and ionic radii; therefore, alkaline conditions favor the dissolution of fluoride minerals. Other geochemical processes, such as ion exchange, arid climates with high evaporation rates, mixing, and calcite precipitation also promote the fluoride enrichment in groundwater [17,18]. In addition to these natural sources, anthropogenic inputs including agricultural irrigation, phosphate fertilizer application, and brick kilns also play a role on fluoride concentration in groundwater [19,20]. Endemic fluorosis is prevalent in rural area because of the limitation of social economy. Absence of treatment procedures elevates the exposure to high fluoride drinking water.

In China, extensive groundwater fluoride contamination is primarily observed in northern China largely resulting from dissolution of F-bearing minerals. In northwestern arid inland basins, such as Manas River Basin, Zhangye Basin, Hetao Plain, Datong Basin, Taiyuan Basin, Yuncheng Basin, Guanzhong Basin, and the Huhhot Basin, all of which are controlled by continental arid and semi-arid climates, high fluoride groundwaters largely exist in Quaternary lacustrine and fluvial aquifers [21,22,23,24,25,26,27]. Fluorite dissolution in sediments and accumulation through water–rock interactions are the primary mechanisms attributed to F enrichment in these areas. Evaporation further promotes increases in groundwater fluoride ion concentrations. For example, the maximum value of fluoride in shallow groundwaters peaked at 22 mg/L in the discharge or evaporation zone of the Datong basin [28]. Generally, F concentrations in shallow groundwaters are higher than deep groundwater levels due to evaporation. Previous research has shown that large-scale groundwater fluoride contamination is regarded as a common concern in northern China from the west Manas Basin through HeXi Corridor and Guanzhong Basin to the eastern coastal areas. 

Located in the fluoride belt zone, this study area is controlled by regional hydrogeology and climate conditions, but with respect to the local residents, knowledge of spatial distribution and influential factors of fluoride is crucial to ensure safe groundwater supplies. The study area is located along the northern edge of the Guanzhong basin and connected to the Chinese Loess Plateau. The loess aquifer distributes widely in the study area and has some unique features. Loess is rich in voids and fractures that act as both transport paths and water-containing spaces. Furthermore, the loess contains various clay minerals and chemical compositions. Consequently, the loess aquifer presents a spatially heterogeneous structure and the water–rock interaction is a slow, yet complex process that could have significant influence on dissolution of fluoride minerals [29,30,31]. In this study, we found that domestic drinking water was largely derived from both public and private supply wells. Therefore, groundwater quality significantly impacts human health. The problem of groundwater fluoride in this loess aquifer, however, has not been investigated comprehensively. Therefore, the purpose of this study is to (1) determine the diversity of fluoride occurrence in loess aquifer and adjacent aquifers, (2) elucidate the correlation between fluoride enrichment and geochemical environment, (3) assess the human health risk induced by high fluoride concentration in drinking water, and (4) discuss the management dilemma and possible options of drinking water safety strategies in rural areas.

## 2. Materials and Methods 

### 2.1. Study Area Description

The study area (3425 km^2^) is located along the southern edge of the Chinese Loess Plateau, approximately 40 km northwest from Xi’an. The continental monsoon, semi-humid climate has a mean annual air temperature of 12.7 °C and precipitation of 546 mm. The rainy season usually occurs from June to September. The Qishui River and Jing River form the west and east boundaries. The Mogu River is a tributary of the Qishui River and flows from north to southwest; the Gan River is a tributary of the Jing River and flows north to southeast. The easterly flowing Wei River forms the southern boundary of the study area. This area is characterized by both hilly and plain landscapes with altitudes ranging from 370 to 1600 m (Figure 1).

The study area falls within the transition zone of the southern edge of the Chinese Loess Plateau and Guanzhong Basin. It is limited to the north by the Bei Mountains, with the three rivers for the other boundaries. Therefore, this watershed is mainly composed of three landforms; a hilly area in the north, with a loess tableland and an alluvial plain successively distributed in the south. 

The geological formations in the hilly areas are dominated by sedimentary rocks. The strata mostly consist of Sinian limestone, Cambrian–Ordovician limestone, Permian–Triassic quartz sandstone, mica shale, and mudstone. The remarkable geological feature of the southern plain area is the overlying Quaternary loess whose thickness varies from 30–150 m and is the main unconfined aquifer (Figure 2). The top layer of the loess (upper Pleistocene loess) is distributed widely across the tableland and triple terrace with a thickness of 10–20 m. Located 10–150 m below the surface, the middle Pleistocene loess is the main phreatic aquifer in loess tableland. Quaternary alluvium and diluvium are widely distributed underlying the loess and consists of the unconfined aquifer. 

Groundwater was generally found in four aquifer types: (1) The upper Quaternary loess aquifer (I) located in middle and southern area is the primary water source for private wells because the shallow depth (< 80 m deep) makes drilling economical. The depths of well are usually less than 80 m. It is commonly used for domestic water and standby drinking water. (2) The lower alluvial aquifer (II) under the loess tableland is utilized for public supply wells and irrigation wells (depths 100–300 m). Because of shallow aquifer overexploitation, which eventually fails to satisfy water demand, sufficient and reliable drinking and irrigation water come mainly from the confined aquifer. (3) The alluvial aquifer (III), an abundant water source on the alluvial plain of the Wei River, consists of phreatic water, shallow confined water, and deep confined water. (4) Fracture water found in bedrock aquifer (IV; well depths > 300 m) in the northern hilly area is the most significant water company groundwater source site, which supplies water for urban use.

### 2.2. Sample Collection and Laboratory Analysis

During July 2018, a sampling program was conducted to investigate the hydrochemical characteristics of the study area groundwater. In total, 126 samples were collected from wells used for drinking.

At each sampling site, temperature, pH, and EC were measured in situ with a portable meter (Aquaread AP-700). Prior to collecting, the well was flushed with at least three times the water volume of the tube and the polyethylene sampling bottle was also flushed at least three times prior to filling the 2.5 L bottles. Once the sample was collected, the bottles were immediately sealed and labeled with number, date, water type, air and water temperatures, as well as chemical preservatives. All samples were stored in a cooler and transported to the laboratory within three days. 

All samples were analyzed at the *Water and Soil Testing Center of Shaanxi Institute of Engineering Prospecting* (*SIEP*) and the analysis methods were in accordance with approved, standard testing methods. Fluoride was measured using the ion selective electrode method with the detection limit of 0.2 mg/L. Potassium and sodium were measured using flame atomic absorption spectrometry. Ca^2+^, Mg^2+^, Cl^−^, SO_4_^2−^, and HCO_3_^−^ were analyzed by titration. NO_3_^−^ was analyzed using the thymol spectrophotometry method (detection limit 2.5 mg/L). QA/QC were performed by making duplicate measurements. Charge balances were checked to ensure the accuracy of analysis and errors were within the acceptable limit of ±5%. 

### 2.3. Human Health Risk Assessment

Human health risk assessment was conducted to determine any potential adverse effects. This study utilized the assessment model recommended by USEPA for determining the potential dose through drinking water intake (the primary method of introducing fluoride/other pollutants into the body) [32]. The average daily intake of pollutants in drinking water was calculated using Equation (1): (1)$$ADI_{i}=\frac{C_{i}\times IR\times EF\times ED}{BW\times AT}$$
where *ADI_i_* represents the average daily intake of *i*th pollutant through drinking water intake (mg/kg·day); *C_i_* is the concentration of *i*th pollutant in groundwater (mg/L); *IR* stands for the drinking water ingestion rate, set as 1.5 L/d for adults and 0.7 L/d for children; *EF* represents exposure frequency as 365 days/year. For the non-carcinogenic risk exposure assessment, *ED* is the exposure duration as 30 years for adults and 12 years for children, and the averaging time during exposure (*AT*) values were 10,950 and 4380 days for adults and children, respectively [33]. 

The hazard quotient of non-carcinogenic risk for *i*th pollutant through drinking water intake pathway was determined using Equation (2), where *RfD_i_* represents the reference dose of *i*th pollutant through drinking water intake pathway. The *RfD* value for F^−^ is 0.04 mg/kg/day [34,35].
(2)$$HQ_{i}=\frac{ADI_{i}}{RfD_{i}}$$

The hazard index (*HI*) represents the total non-carcinogenic risk to humans and is calculated using Equation (3). The non-carcinogenic risk is accessible when *HI* < 1, while there is inaccessible high health risk to humans when *HI* > 1.
(3)$$HI=\sum_{i=1}^{n}HQ_{i}$$

## 3. Results and Discussion

### 3.1. Groundwater Fluoride Contamination and Drinking Water Quality 

Groundwater in the study area is slightly alkaline (pH 7.4–8.7; mean 8.1). The predominate cation in groundwater is Na^+^, followed by Mg^2+^. For anions, groundwater is dominated by HCO_3_^−^ and SO_4_^2−^. Figure 3 shows the ion concentration varieties in the four water types. The F^−^ concentration is shown in Figure 4 in addition to the spatial distribution in different aquifers. Hydrochemical facies for all groundwater samples are plotted on Chadha’s plot and shown in Figure 5.

Type I water from loess aquifer is characterized as Na-rich/Ca-poor; the Na^+^ concentration ranges from 29.5 to 391.0 mg/L while the Ca^2+^ concentration ranges from 10.0 to 92.2 mg/L. The excess of Na^+^ is pervasive, and 46% samples of type I water exceeded WHO guidelines. Type I water has a wide range of TDS (Total Dissolved Solids) values (348 to 2188 mg/L, average 861 mg/L). The TDS enrichment is significant, though it only occurs in a fraction (28%) of water samples. For Cl^−^ and SO_4_^2−^, although the maximum values are 1.5 and 2 times WHO limits, respectively, a majority of samples (96%) are within the limits. The mean value of NO_3_^−^ in groundwater significantly exceeds the WHO limit of 50 mg/L for drinking water, with 36% of groundwater samples exceeding this limit with a maximum of 13 times the WHO limit. The highest F^−^ values were observed in loess aquifer and ranged from 0.76 to 3.80 mg/L (mean 1.74 mg/L) with 64% of water samples exceeding the limit of 1.5 mg/L. In general, water quality issues in type I water were fluoride pollution as well as Na^+^ and NO_3_^−^ enrichment. 

Type II water from the lower alluvial aquifer underlying the loess has lower salinity as compared to type I water. TDS values range from 468 to 1240 mg/L (mean 786 mg/L) with Na^+^ still the dominant (107–343 mg/L), and 60% of samples exceeding WHO guidelines. Of the anions, only Cl^−^, SO_4_^2−^, and NO_3_^−^ exceed WHO guidelines in only 11%, 7%, and 23% of samples, respectively. The F^−^ concentrations vary from 0.76 to 3.02 mg/L (mean 1.41 mg/L). Fluoride pollution is alleviated in type II water, of which 30% of samples have F values that exceed 1.5 mg/L. Generally, type II water quality is superior to type I, which probably due to less anthropocentric impacts. 

Type III water from the sandy gravel aquifer is distributed along Wei River, the groundwater discharge area, presenting higher Ca^2+^, Mg^2+^, Cl^−^, SO_4_^2−^, and TDS values. TDS values exceed a remarkable 3580 mg/L in the southeast discharge area; the highest levels of SO_4_^2^^−^ and Cl^−^ were 1311 mg/L and 539 mg/L, respectively. Type III water had the lowest F^−^ content (0.38–1.35 mg/L, mean 0.87 mg/L). 

Type IV water from the fracture aquifer is low salinity with TDS values varying between 304 and 612 mg/L. The F^−^ concentration ranges from 0.59 to 1.48 mg/L (mean 1.02 mg/L). Except for one sample collected in a shallow well, all samples from deep wells are within WHO limits. 

In general, it is notable that most samples from type I, type II, and type III water significantly exceed WHO guidelines for Na^+^, Cl^−^, SO_4_^2−^, NO_3_^−^, F^−^, and TDS. Type IV water in the hilly area has superior drinking water quality. 

To better display the hydrochemical classification of the groundwater samples, the difference in milliequivalent percentage between alkaline earths (Ca + Mg) and alkali metals (Na + K) is plotted against the difference in milliequivalent percentage between weak acidic anions (CO_3_ + HCO_3_) and strong acidic anions (Cl + SO_4_) [36]. As shown in Figure 5, most groundwater samples that contain high concentrations of fluoride are Na–HCO_3_ type, Na-mixed anions type, and mixed cations-HCO_3_ type waters; these samples largely occur in type I and type II waters. In the hilly northern area, Na–HCO_3_ and Ca·Mg–HCO_3_ represent the primary chemical facies of type IV water. The slow movement of groundwater from the hilly area to the alluvial plain of the Wei river promotes rock–water interactions (solute dissolution and cation exchange) with Ca·Mg–HCO_3_ type, mixed cations-HCO_3_ type, and Ca·Mg-mixed anions type waters becoming dominant in type III waters. 

### 3.2. Effect of Hydrogeochemical Processes

#### 3.2.1. Fluoride-Containing Mineral Dissolution

Fluoride in groundwater is normally associated with fluoride-containing mineral dissolution. The primary mineral sources of fluoride include biotite, fluorite, and amphiboles, which release F^−^ into groundwater under alkaline conditions [37]. Equations (4) and (5) show fluoride ion displacement from minerals (biotite and amphiboles) by hydroxide. Dissolution of fluorite (Equations (6) and (7)) is the most common source of fluoride in groundwater [38].
(4)$$KMg_{3}\left[AlSi_{3}O_{10}\right]F_{2}+2OH^{-}\to KMg_{3}\left[AlSi_{3}O_{10}\right]{\left[OH\right]}_{2}+2F^{-}$$
(5)$$NaCa_{2}{\left[Mg,Fe,Al\right]}_{5}{\left[Al,Si\right]}_{8}O_{22}F_{2}+2OH^{-}\to NaCa_{2}{\left[Mg,Fe,Al\right]}_{5}{\left[Al,Si\right]}_{8}O_{22}{\left[OH\right]}_{2}+2F^{-}$$
(6)$$CaF_{2}+2NaHCO_{3}\to CaCO_{3}+2F^{-}+2Na^{+}+H_{2}O+CO_{2}$$
(7)$$CaF_{2}+H_{2}O+CO_{2}\left(g\right)\to CaCO_{3}+2F^{-}+2H^{+}$$

As shown in Figure 6b, saturation index (SI) values of fluorite in all groundwater samples range from −1.8 to −0.5, indicating the fluorite is undersaturated with respect to groundwater. Therefore, dissolution of fluoride is a major source for the enrichment of F^−^ in groundwater. 

#### 3.2.2. pH Impacts

The pH of groundwater significantly impacts the dissolution of fluorite-bearing minerals. Figure 6a shows high fluoride ion concentrations in type I and type II water as well as high pH values; the loess aquifer contains abundant amounts of exchangeable F^−^ and alkaline condition promote the release of F^−^. Type III water has low F^−^ concentrations and low pH values; type IV water has low F^−^ concentrations but high pH values. The sample in type IV water with high F^−^ concentration was collected from a shallow well where the aquifer is covered by loess. Therefore, one conclusion drawn from this result is that alkaline condition promotes fluoride ion release because of OH^−^/F^−^ exchange in F^−^-rich minerals.

#### 3.2.3. Calcium Removal Processes

Equation (8) shows that Ca^2+^ influences fluorite dissolution, as well as the processes causing Ca^2+^ variation. As the bivariate plot shows, F^−^ is negatively related to Ca^2+^ (Figure 6c), indicating calcium removal promotes fluorite dissolution and results in groundwater fluoride enrichment [39]. Calcium removal likely involves cation exchange and calcite precipitation. 

Cation exchange in groundwater involves replacement of Ca^2+^ and Mg^2+^ by Na^+^ in minerals, which varies the chemical compositions of groundwater. The process is evaluated by the relation between two parameters (Na–Cl and Ca + Mg–SO_4_–HCO_3_). Na–Cl represents the amount of Na gained or lost from sources except halite, whereas Ca + Mg–SO_4_–HCO_3_ represents the amount of Ca and Mg gained or lost relative to that provided by dissolution of carbonate and gypsum. According to Equation (8), if cation exchange is the dominant process of Ca^2+^, Mg^2+^, and Na^+^ variations, the relationship between these two parameters should be linear with the slope of −1.0 [40]. As shown in Figure 7a, all groundwater samples fit a line with a slope of −1.02 (*r*^2^ = 0.88), indicating cation exchange is responsible for the consumption of Ca^2+^ in groundwater.
(8)$$\begin{array}{l} Ca^{2+}\\ Mg^{2+} \end{array}\}+2Na-clay=2Na^{+}+\left\{ \begin{array}{l} Ca^{2+}\\ Mg^{2+} \end{array}\right.-clay$$

Cation exchange is further confirmed by examining chloro-alkaline indices (CAI-1 and CAI-2). Almost all groundwater samples in this study fall into the negative zone, indicating the occurrence of cation exchange is pervasive in aquifer. Cation exchange impacts chemical equilibrium and subsequent fluoride enrichment. As shown in Figure 7b, the reaction extent and intensity of cation exchange differs in the aquifers. CAI-1 and CAI-2 values are more negative in type I and type II water where higher F^−^ values were observed. In type III, low-fluoride water from the alluvial plain, CAI-1 and CAI-2 values tend to be positive, indicating that cation exchange is insignificant. 

Ca^2+^ could also be reduced by precipitation. The saturation index (SI) helps determine the stability of minerals with respect to groundwater and indicates the tendency of dissolution or precipitation. The SI for this work was calculated using PHREEQC software. According to mineral SI values, the groundwater is saturated with dolomite and calcite while undersaturated with fluoride and gypsum (Figure 8a,b,e). Dolomite and calcite are the primary sources of Ca^2+^ and Mg^2+^ in groundwater. Dolomite and calcite tend to dissolve as calcium ions are removed. The Ca/Mg ratio is used to determine the dominate dissolution of calcite and dolomite. When the Ca/Mg ratio >1, calcite dissolution predominates, while a Ca/Mg ratio <1 indicates dolomite dissolution is the dominant process. Most samples have the Ca/Mg ratio >1, shown in Figure 8c, and indicate that groundwater from hilly areas and the loess tableland area where the upstream flow path primarily contains dolomite. Dolomite dissolves slowly along the flow path, which inevitably results in calcite oversaturation and precipitation. Dissolution of gypsum is another influential process regarding calcite precipitation. SI values of gypsum are plotted against calcite and dolomite. Groundwater is undersaturated with gypsum but oversaturated with calcite and dolomite (Figure 8a,b). 

It is noticeable that the fluoride concentration in groundwater is largely controlled by the dissolution of F-containing minerals. SI values of fluorite vary significantly with Ca^2+^ concentration (Figure 8f). Therefore, gypsum dissolution and calcite precipitation influence the dissolution of F-containing minerals by controlling Ca^2+^ concentration [41,42]. 

#### 3.2.4. Residence Time of Groundwater in Aquifer and Potential Evaporation Impacts

The stable isotopes of hydrogen and oxygen in water (δD and δ^18^O) provide information on water origins, subsequent evaporative processes, and groundwater residence time. Stable isotopes do not provide the groundwater age, but they do indicate the climatic conditions when groundwater is recharged [43]. Figure 9a shows the values of δD and δ^18^O obtained from groundwater samples. Fluoride ion concentrations are plotted against δ^18^O in Figure 9b. The values of δD and δ^18^O of groundwater are scattered with a wide range (−48.05‰ to −86.12‰ and −5.98‰ to −11.61‰, respectively). All groundwater samples plot to the right of the local meteoric water line (LMWL) with a parallel trend, indicating groundwater in the study area was recharged under different climate conditions in the past and subjected to some degree of evaporation during infiltration [44,45]. The offset of groundwater isotope values from LMWL indicate the groundwater is old; the residence time of groundwater in the aquifer is long and the groundwater flow is slow. Long residence times induce sufficient water–rock interactions that play a significant role on the resolution of fluoride minerals. 

Shallow groundwater aquifers (type I and III) show higher stable isotopic values than deep groundwaters (types II and IV) except one type IV outlier, which was collected in a shallow well. Stable isotopes of type I water are plotted linearly with the slope of 5.47, indicating the shallow groundwater in the loess aquifer is obviously subjected to evaporation. As shown in Figure 9b, type I water is characterized by high fluoride levels coupled with high δ^18^O values. Type III water (alluvial aquifer) is closer to LMWL with a slope of 7.18, with low fluoride levels and high δ^18^O values, indicating good circulation between groundwater and modern meteoric precipitation. Low fluoride levels in alluvial aquifers are associated with active runoff conditions. Type II water occurs in a deep alluvial aquifer. There are a few samples from type II water that show low stable isotopic values and enriched fluoride concentrations, which are indicators of inter-aquifer flow with the loess aquifer. Most type II and IV waters from deep aquifers have low isotopic and fluoride values. These findings suggest that enrichment of fluoride in groundwater is controlled by lithology of aquifer and cycle conditions. Particular characteristics of loess strongly favor enrichment of fluoride in groundwater. 

### 3.3. Effect of Anthropogenic Activities

Phosphate-containing fertilizers usually serve as an additional groundwater fluoride source. Some phosphatic fertilizers including superphosphate (2750 mg of F^−^/kg), potash (10 mg of F^−^/kg), and NPK (Nitrogen Phosphorous Potassium) (1675 mg of F^−^/kg) contain large amounts of fluoride. Strong positive correlations between fluoride and nitrate have been previously reported and illustrate the influence of fertilizers on groundwater fluoride enrichment. Cultivated land within the study area accounts for 80% of the total area; this implies that tillage is the main anthropogenic activity. As shown in Figure 6g, there is no significant correlation between fluoride and nitrate, indicating the impact of agricultural activities on groundwater fluoride contamination are negligible. 

### 3.4. Human Health Risk Assessment

The relationship between the high fluoride geochemical environment and endemic fluorosis is quite clear. Generations from the 1960s to the 1990s in the study area generally suffered from readily detected dental fluorosis. Elevated fluoride concentrations in the groundwaters of this study possibly pose a non-carcinogenic risk to human health via oral intake. Therefore, a human health risk assessment was conducted according to the aforementioned method. Figure 10 illustrates the spatial distribution of risk for adults and children. Maximum values of *HI* were observed in type I water (2.5 for adults, 4.2 for children). The mean values of HI from the four water groups (Types I–IV) were 1.3, 0.9, 0.6, 0.8 and 2.1, 1.6, 1.0, 1.3 for adults and children, respectively. 

In type I water, 60% of samples had HI values >1 for adults, indicating most groundwater in the loess area probably poses a non-carcinogenic risk to human health. The risk is more significant for children, as 94% of type I water samples had *HI* values > 1. In type II water, there were 23% and 73% samples with *HI* values > 1 for adults and children, respectively. The risk in types III and IV water was insignificant for adults, though 52% and 70% of samples posed potential health risks for children. In general, the human health risk assessment indicated that elevated fluoride levels in groundwater would significantly cause a non-carcinogenic risk for both adults and children, with a particular emphasis on children, who are more vulnerable. 

Although there are some default assumptions, limitations, and uncertainties resulting from a conservative and underestimated assessment, the present study disclosed an exposure risk to drinking water with high fluoride ion concentration levels and provides a clarified knowledge of the possible human health risk for policy-makers in this arid loess area. 

### 3.5. Dilemma of Drinking Water Management and Possible Options

Groundwater containing elevated levels of fluoride ion are typical in rural areas where water treatment procedures are absent. Numerous fluoride removal methods have been introduced to decrease fluoride levels in drinking water [46]. As early as the 1980s, reverse osmosis was proposed as an effective method for reducing fluoride ion concentration for small water systems. Other methods were subsequently introduced, including electro-dialysis, ion exchange, nanofiltration, limestone reactor, and activated alumina [47]. However, social and economic concerns limited the application of these methods in remote areas [48]. Chronic fluoride ingestion results in an endemic disease in which the risk to human health has already materialized and is irreversible. 

This study area featured drinking water supplies that consisted of two patterns; one is a public water supply project funded by the government, which withdraws water from deep wells to supply adequate levels of community water; the other is private shallow wells used to satisfy the household water demand. Both patterns face some challenges. This work shows that type II and IV waters from deep wells (the main source for public wells) have superior quality as compared to shallow wells. However, due to the hydraulic connection between aquifers, deep groundwater is gradually influenced by contaminated shallow groundwaters. Furthermore, vertical fractures and root holes have developed in the loess aquifer, which favor the preferential flow and increases the contamination risk of the deep aquifer. Development of deep groundwaters without considering shallow groundwater quality concerns is not a sustainable pattern. Furthermore, with respect to small communities in rural area, groundwater withdrawn from a shallow aquifer is distributed to every household without any treatment. Although tap water treatment equipment has reached 80% in the study area, the practical usage of tap water for drinking is far lower. Villagers regard water as free and are reluctant to pay for it, meaning inferior private shallow wells are still the main source of drinking water. Recently, many villagers have installed household water filters; however, they are unaware of the need to regularly replace the filter. This dilemma is the most common concern in most rural areas of northern China. Therefore, it is imperative for the government to take some effective measures to tackle this problem. Fluoride removal techniques and optimizing water supply strategies are possible options for drinking water safety management. It works quickly to supply low fluoride drinking water by public wells. The combination of optimizing water supply strategy and popularizing fluoride removal techniques such as reverse osmosis is a feasible long-term option. 

## 4. Conclusions

Fluoride contamination of aquifers along the south edge of the Chinese Loess Plateau in west-central China was investigated using a combination of hydrochemistry and stable environmental isotopes. The diversity of fluoride occurrence in the loess and adjacent aquifers elucidated a correlation of fluoride concentration with the geochemical environment. In the shallow loess aquifer, high concentrations (0.76 to 3.8 mg/L, mean 1.74 mg/L) of fluoride ions were observed in the groundwater as well as δ^18^O values (−10.29‰ to −7.25‰); in the downstream shallow alluvial aquifer, groundwater had low fluoride levels (0.38 to 1.35 mg/L, mean 0.87 mg/L) and high δ^18^O values (−10.65‰ to 7.00‰). In deep aquifers, both the fracture aquifer and the deep alluvial aquifer had medium fluoride levels (0.59 to 1.48 mg/L, mean 1.02 mg/L and 0.76 to 3.02 mg/L, mean 1.41 mg/L) and low δ^18^O values (−9.98‰ to −8.36‰, −11.61‰ to −8.81‰). Elevated fluoride levels in shallow loess groundwater comes primarily from F-containing mineral dissolution, which is affected by alkaline condition, calcium-removing processes, groundwater residence time, and evaporation. 

Human health risk assessment data indicated that elevated fluoride levels in shallow loess groundwater (type I water) posed a non-carcinogenic risk on human health for both adults (60% samples *HI* >1) and children (94% samples *HI* >1), with a particular emphasis on children as they are more vulnerable and susceptible to elevated fluoride levels. For the deep fracture groundwater (type IV water), all samples had *HI* < 1 for adults and 70% samples had *HI* >1 for children. For deep alluvial aquifer (type II water), there were 23% and 73% samples with *HI* >1 for adults and children, respectively. 

In addition to fluoride, TDS, Na^+^, Cl^−^, SO_4_^2−^, and NO_3_^−^ of shallow loess groundwater exceeded the WHO limits with the maximum values of 2188.0 mg/L, 391.0 mg/L, 386.0 mg/L, 307.0 mg/L, and 645.0 mg/L, respectively. All the samples from the deep fracture aquifer were within WHO limits. It can be concluded from the above results that deep groundwater has better quality than shallow groundwater. However, due to social and economic limitations, villagers prefer using shallow groundwater from private wells for their drinking water, which pose potential dilemmas for drinking water management in rural areas. In the short term, deep groundwater can be used for domestic water for gradual implementation. In the long run, however, overexploitation of deep aquifers may cause fluoride contamination derived from shallow groundwater leaking. Therefore, for the highly populated loess tableland area, formulating a scientific water supply strategy while considering hydrochemistry and water balance would be both sensible and sustainable.

## Figures and Tables

**Figure 1 ijerph-16-01683-f001:**
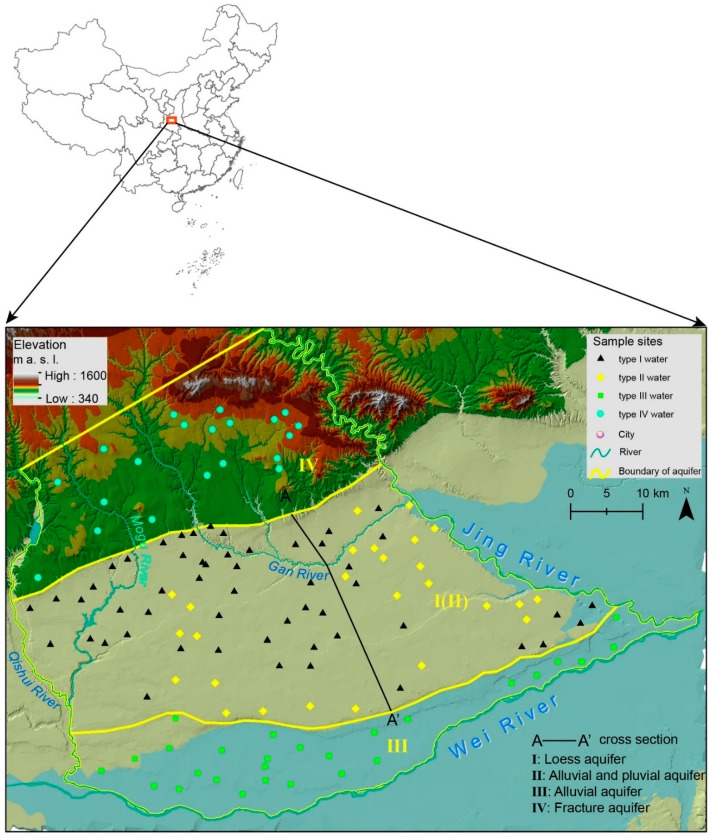
Study area map, showing the terrain relief, rivers, and sample sites.

**Figure 2 ijerph-16-01683-f002:**
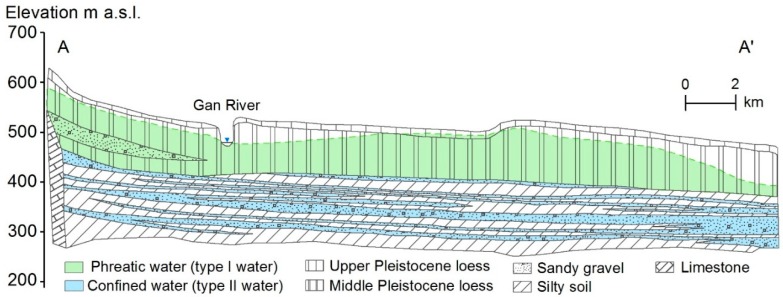
Cross section of loess tableland showing the distribution of phreatic water, confined water, and aquifer lithology.

**Figure 3 ijerph-16-01683-f003:**
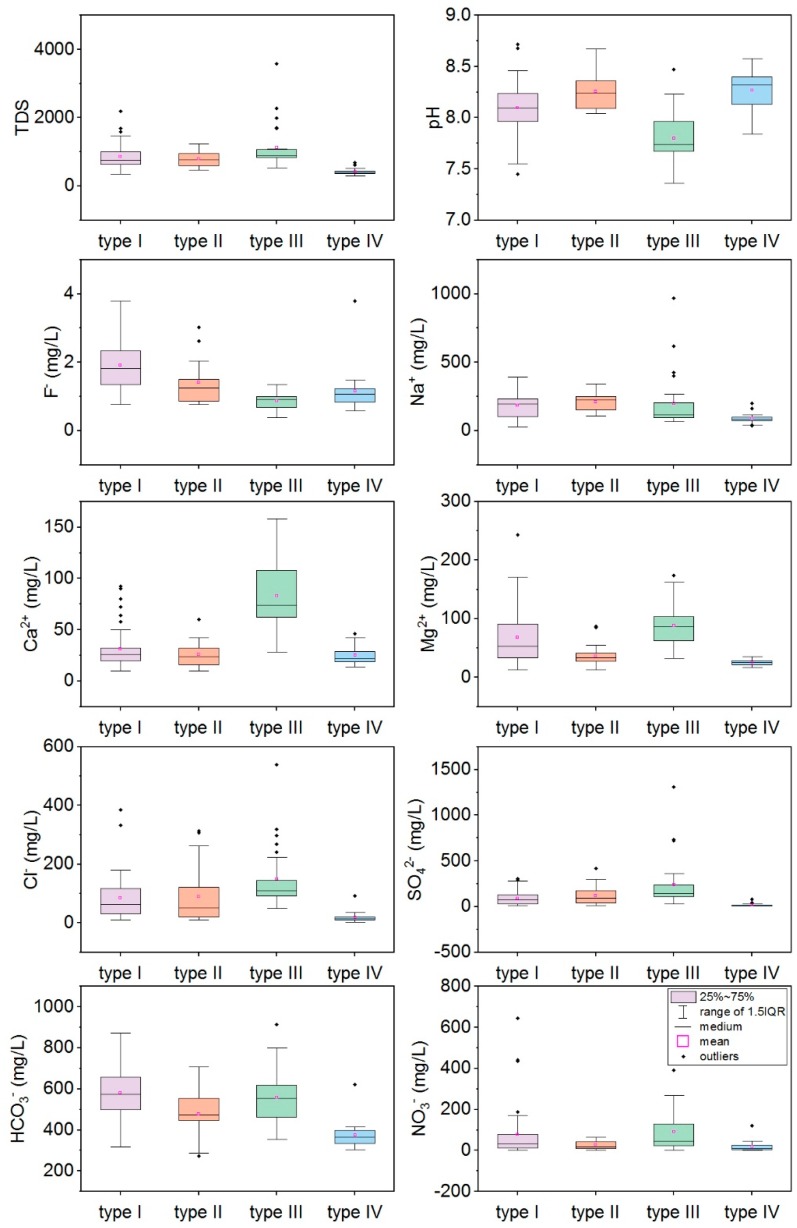
Box plots for the major ions (TDS, pH, F^−^, Na^+^, Ca^2+^, Mg^2+^, Cl^−^, SO_4_^2−^, HCO_3_^−^, NO_3_^−^) in four groundwater aquifers. Type I water came from a shallow loess aquifer, type II water from a deep alluvial aquifer, type III water from a shallow alluvial aquifer, and type IV water from a deep fracture aquifer.

**Figure 4 ijerph-16-01683-f004:**
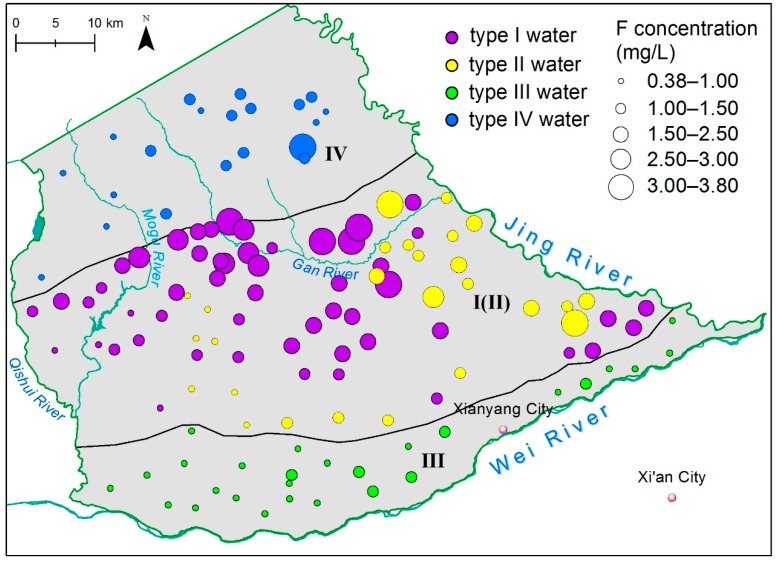
The distribution of fluoride concentration from all groundwater samples.

**Figure 5 ijerph-16-01683-f005:**
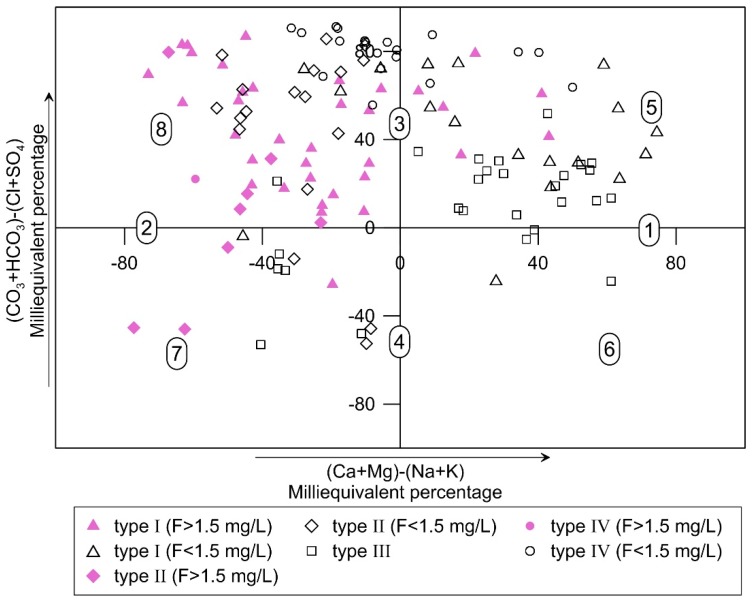
Chadha’s plot showing hydrochemical classification of groundwater. 1. Alkaline earths exceed alkali metals; 2. alkali metals exceed alkaline earths; 3. weak acidic anions exceed strong acidic anions; 4. strong acidic anions exceed weak acidic anions; 5. Ca·Mg–HCO_3_ type, mixed cations-HCO_3_ type, Ca·Mg-mixed anions type waters; 6. Ca·Mg–Cl type, mixed cations -Cl type, Ca·Mg-mixed anions type waters; 7. Na–Cl type, Na-mixed anions type, mixed cations-Cl type waters; 8. Na–HCO_3_ type, Na-mixed anions type, mixed cations-HCO_3_ type waters.

**Figure 6 ijerph-16-01683-f006:**
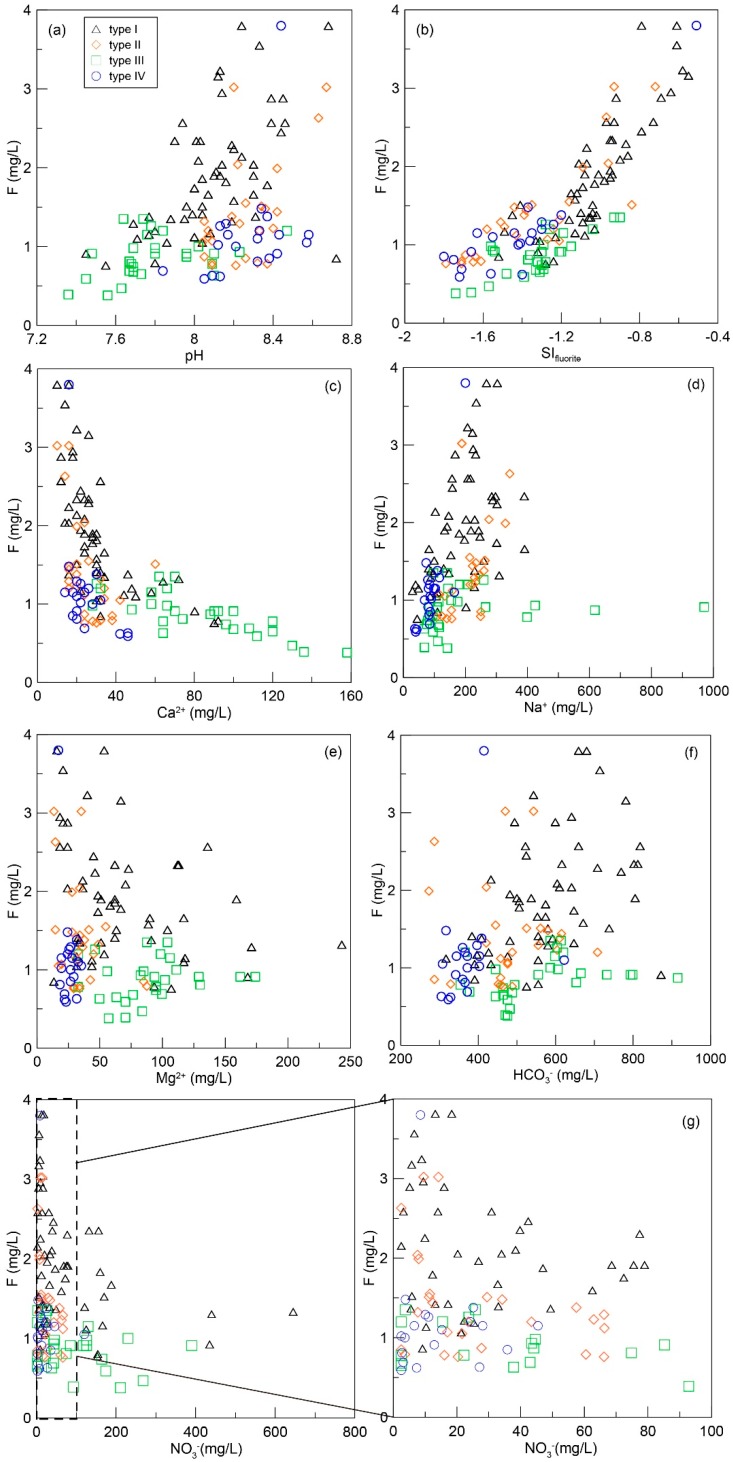
Scatter plots showing fluoride variation with hydrochemistry in groundwater. F^−^ concentration versus (**a**) pH, (**b**) saturation index of fluorite, (**c**) Ca^2+^ concentration, (**d**) Na^+^ concentration, (**e**) Mg^2+^ concentration, (**f**) HCO_3_^−^ concentration, (**g**) NO_3_^−^ concentration.

**Figure 7 ijerph-16-01683-f007:**
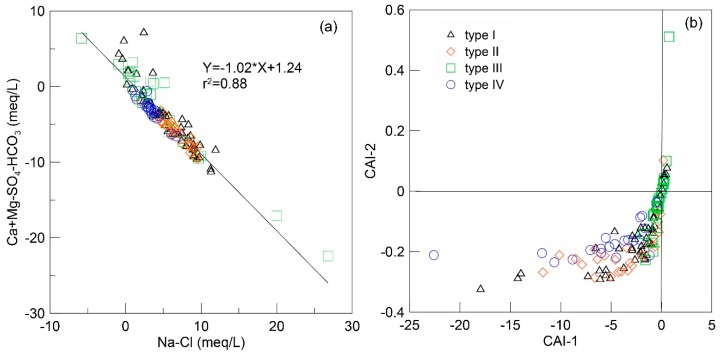
Scatterplots of (**a**) Ca + Mg–SO_4_–HCO_3_ versus Na–Cl and (**b**) chloro-alkaline indices showing the groundwater cation exchange process.

**Figure 8 ijerph-16-01683-f008:**
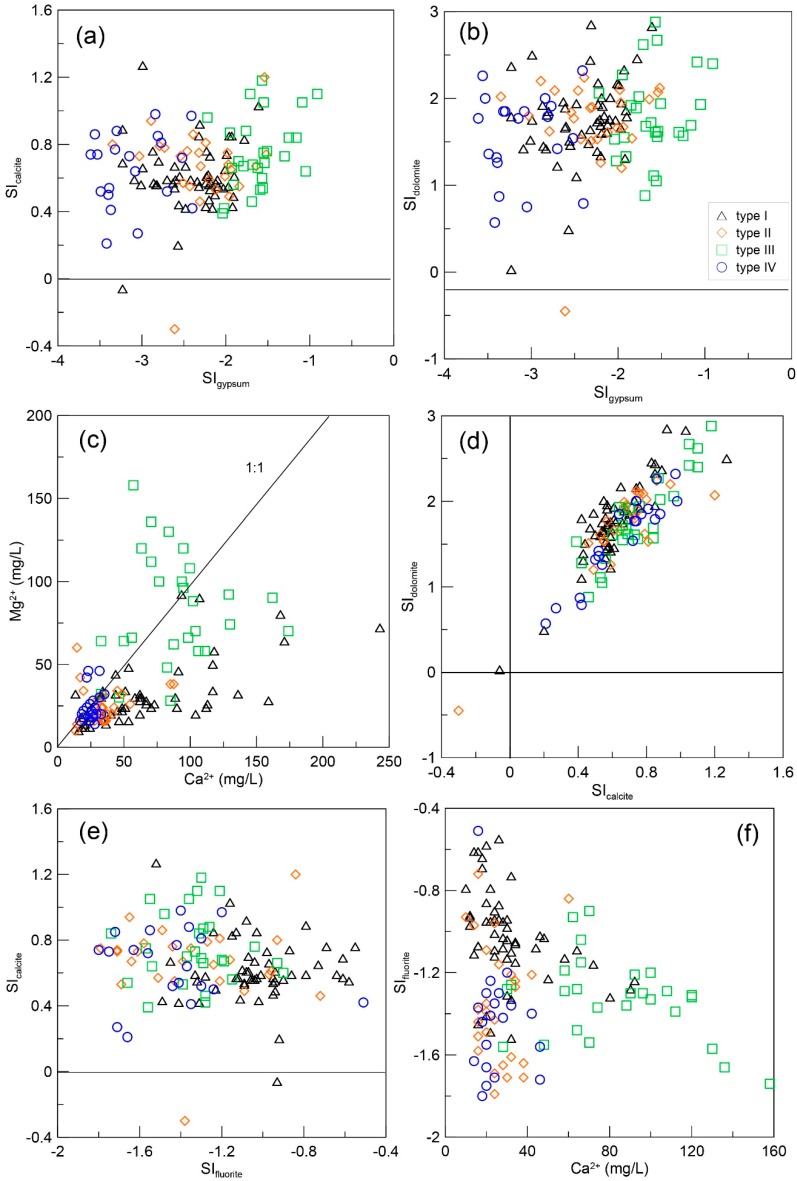
Saturation index (SI) scatterplots of (**a**) calcite, (**b**) dolomite versus gypsum, (**c**) Ca^2+^ versus Mg^2+^, (**d**) SI of dolomite versus calcite, (**e**) SI of calcite versus fluorite, and (**f**) SI of fluorite versus Ca^2+^.

**Figure 9 ijerph-16-01683-f009:**
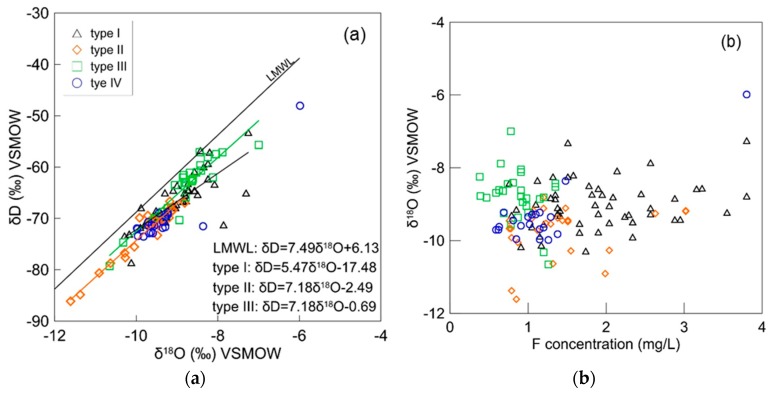
δ^18^O value plotted against (**a**) δD value and (**b**) F^−^ concentration for area groundwater samples.

**Figure 10 ijerph-16-01683-f010:**
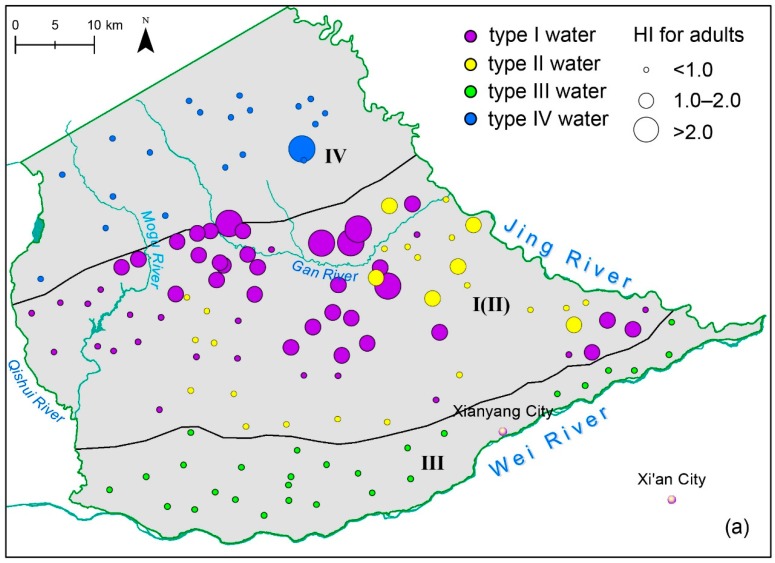
Hazard index (*HI*) of fluoride spatial distributions for (**a**) adults and (**b**) children.

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
