# Peer review of "Fluoride Occurrence and Human Health Risk in Drinking Water Wells from Southern Edge of Chinese Loess Plateau"

_ijerph, 2019, doi:10.3390/ijerph16101683_

Round 1
Reviewer 1 Report
The article is vital to global readers those who are interested in the detrimental effects of fluoride all over the world. The authors have represented a strong abstract which can impress readers at the first glance. The materials and methods section and the results and discussion section of the article were well organized and has clearly discussed the facts with clear figures to give an in-depth analysis of the problem in China but I suggest the authors to address the followings to enhance the quality of the manuscript.
1. In the introduction section, lines 42-48, you have explained about the countries which have recorded high fluoride concentration but you haven’t included Sri Lanka which is also suffering from the high concentration of fluoride. Please include references in that section.
2. I am not quite contented with the conclusion section as it can be implement further to give more detailed overview of the study.
In line 382, it was mentioned “high concentration of fluoride”. Please provide a value or a range of values to specify high concentration rather than just writing high concentration.
In line 383, it was mentioned “low fluoride levels”. Please provide a value or a range of values to specify low fluoride level rather than just writing low fluoride levels.
In line 394, it was mentioned “has better quality”. Please provide a value or a range of values to specify better quality rather than just writing better quality.
Author Response
Dear reviewer,
Thank you so much for your helpful comments and suggestions. We have addressed all the comments carefully and revised our manuscript. The revisions are highlighted in red.
Yours sincerely,
Hui Qian
Point 1: In the introduction section, lines 42-48, you have explained about the countries which have recorded high fluoride concentration but you haven’t included Sri Lanka which is also suffering from the high concentration of fluoride. Please include references in that section.
Response 1: Thanks for the reviewer’s comment. The relevant reference Young, S.M. et al. Environ. Earth Sci. 2011, 63, 1333–1342 has been added. Please see line 38 and number 12 reference.
Point 2: I am not quite contented with the conclusion section as it can be implement further to give more detailed overview of the study.
(1) In line 382, it was mentioned “high concentration of fluoride”. Please provide a value or a range of values to specify high concentration rather than just writing high concentration.
(2) In line 383, it was mentioned “low fluoride levels”. Please provide a value or a range of values to specify low fluoride level rather than just writing low fluoride levels.
(3) In line 394, it was mentioned “has better quality”. Please provide a value or a range of values to specify better quality rather than just writing better quality.
Response 2: Thanks for the reviewer’s comments. We have revised the conclusion section and provided more detailed information.
(1) We have provided the range and mean value (0.76 to 3.8 mg/L, mean 1.74 mg/L) after “high concentrations”, please see line 377.
(2) We have provided the range and mean value (0.38 to 1.35 mg/L, mean 0.87 mg/L) after “low fluoride levels”, please see line 379.
(3) We have provided detailed information to specify the water quality, please see line 385 to 395.
Reviewer 2 Report
It has long been known that fluoride may be dangerous to human health and its application must be strictly controlled. Taking this into account, one should notice that the level of novelty of this research article is low. On the other hand, the results and conclusions of the paper may be interesting for services responsible for public health in China.
The most controversial are statements appearing in the introduction. The authors claim that "Fluoride is an essential trace element for the body with important physiological functions".
I totally disagree with this statement. Fluoride is not and never has been essential for the body and people have lived without fluoride for ages. Under certain conditions it may be beneficial, but "beneficial" and "essential" does not mean the same. Antibiotics are beneficial and helpful in the treatment of infectious diseases caused by bacteria but they are obviously not essential for human body.
Moreover, what kind of "important physiological functions" of fluoride have the authors in mind? Could they mention any of them? Anticariogenic effect of fluoride cannot be considered as "physiological function". Accumulation of fluoride in hard tissues is a response of the body to the exposure to exogenous substance and nothing more. Physiologic functions of fluoride are simply unknown. On the contrary, we know many harmful actions of fluoride in humans and other living organisms e.g. inhibition of regulatory enzymes involved in the key biochemical pathways.
The authors are of the opinion that "Fluoride in drinking water below 0.5 mg/L causes dental caries". This is a nonsense. Caries is a breakdown of teeth due to the action of organic acids produced by bacteria in the dental plaque and fluoride has nothing to do with its pathogenesis. However, under certain conditions it may be useful in prevention of this pathology by inhibiting demineralization and stimulating remineralization of enamel as well as by inhibiting bacterial enzymes involved in anaerobic glycolysis in which organic acids are produced.
In my opinion, the above mentioned misleading statements must be removed from the manuscript.
Manuscript should be checked for spelling mistakes e.g. on page 2 in line 71: "primiary".
Colors used in Figure 10 could be more diverse.
Author Response
Dear reviewer,
Thank you so much for your helpful comments and suggestions. We have addressed all the comments carefully and revised our manuscript. The revisions are highlighted in red.
Yours sincerely,
Hui Qian
It has long been known that fluoride may be dangerous to human health and its application must be strictly controlled. Taking this into account, one should notice that the level of novelty of this research article is low. On the other hand, the results and conclusions of the paper may be interesting for services responsible for public health in China.
Point 1: The most controversial are statements appearing in the introduction. The authors claim that "Fluoride is an essential trace element for the body with important physiological functions".
I totally disagree with this statement. Fluoride is not and never has been essential for the body and people have lived without fluoride for ages. Under certain conditions it may be beneficial, but "beneficial" and "essential" does not mean the same. Antibiotics are beneficial and helpful in the treatment of infectious diseases caused by bacteria but they are obviously not essential for human body.
Moreover, what kind of "important physiological functions" of fluoride have the authors in mind? Could they mention any of them? Anticariogenic effect of fluoride cannot be considered as "physiological function". Accumulation of fluoride in hard tissues is a response of the body to the exposure to exogenous substance and nothing more. Physiologic functions of fluoride are simply unknown. On the contrary, we know many harmful actions of fluoride in humans and other living organisms e.g. inhibition of regulatory enzymes involved in the key biochemical pathways.
The authors are of the opinion that "Fluoride in drinking water below 0.5 mg/L causes dental caries". This is a nonsense. Caries is a breakdown of teeth due to the action of organic acids produced by bacteria in the dental plaque and fluoride has nothing to do with its pathogenesis. However, under certain conditions it may be useful in prevention of this pathology by inhibiting demineralization and stimulating remineralization of enamel as well as by inhibiting bacterial enzymes involved in anaerobic glycolysis in which organic acids are produced.
In my opinion, the above mentioned misleading statements must be removed from the manuscript.
Response 1: Thanks for the reviewer’s comments and detailed interpretation. We have read relevant references carefully, and realized the statements of fluoride in the introduction section are indeed misleading and inaccurate. We have removed the misleading statements. Please see line 33 to 35.
Point 2: Manuscript should be checked for spelling mistakes e.g. on page 2 in line 71: "primiary".
Response 2: Thanks for highlighting the mistake. We have corrected the spelling mistake. Please see line 65.
Point 3: Colors used in Figure 10 could be more diverse.
Response 3: Thanks for the reviewer’s comment. The colors in figure 10 have been modified.
Round 2
Reviewer 2 Report
The article has been improved enough to be qualified for publication in the International Journal of Environmental Research and Public Health.